# Impact of Physical Activity on Disability Risk in Elderly Patients Hospitalized for Mild Acute Diverticulitis and Diverticular Bleeding Undergone Conservative Management

**DOI:** 10.3390/medicina57040360

**Published:** 2021-04-08

**Authors:** Aldo Rocca, Maria Chiara Brunese, Micaela Cappuccio, Andrea Scacchi, Gennaro Martucci, Antonio Buondonno, Fabio Massimo Perrotta, Gennaro Quarto, Pasquale Avella, Bruno Amato

**Affiliations:** 1Department of Medicine and Health Sciences “V. Tiberio”, University of Molise, Via F. Desaanctis 1, 86100 Campobasso, Italy; mariachiarabrunese@libero.it (M.C.B.); micaelacappuccio24@gmail.com (M.C.); scacchiandrea@me.com (A.S.); gennaro.martucci@asrem.org (G.M.); a.buondonno@gmail.com (A.B.); fperrotta@unimol.it (F.M.P.); avella.p@libero.it (P.A.); 2Department of Clinical Medicine and Surgery, University of Naples “Federico II”, Via S. Pansini, 5, 80131 Naples, Italy; gennaro.quarto@unina.it; 3Department of Public Health, University of Naples “Federico II”, Via S. Pansini, 5, 80131 Naples, Italy; bramato@unina.it

**Keywords:** physical activity, disability, elderly, surgery

## Abstract

*Background and Objectives:* The role of physical activity (PA) in elderly patients admitted to surgical units for mild acute diverticulitis in the development of disability has not been clarified so far. Our aim is to demonstrate the relationship between physical activity and better post-discharge outcomes on disability in elderly population affected by diverticular disease. *Materials and Methods:* We retrospectively reviewed data of 56 patients (32 Males-24 females) collected from October 2018 and March 2020 at Cardarelli Hospital in Campobasso. We included patients older than 65 yrs admitted for acute bleeding and acute diverticulitis stage ≤II, characterized by a good independence status, without cognitive impairment and low risk of immobilization, as evaluated by activity of daily living (ADL) and the instrumental activity of daily living (IADL) and Exton-Smith Scale. “Physical Activity Scale for the Elderly” (PASE) Score evaluated PA prior to admission and at first check up visit. *Results:* 30.4% of patients presented a good PA, 46.4% showed moderate PA and 23.2% a low PA score. A progressive reduction in ADL and IADL score was associated with lower physical activity (*p* value = 0.0038 and 0.0017). We consider cognitive performance reduction with a cut off of loss of more than 5 points in Short Port of ADL and IADL and a loss of more than 15 points on Exton-Smith Scale, (*p*-value 0.017 and 0.010). In the logistic regression analysis, which evaluated the independent role of PASE in disability development, statistical significance was not reached, showing an Odds Ratio of 0.51 95% CI 0.25–1.03 *p* value 0.062. *Discussion:* Reduced physical activity in everyday life in elderly is associated with increased post-hospitalization disability regarding independence, cognitive performance and immobilization. *Conclusions:* Poor physical performance diagnosis may allow to perform a standardized multidimensional protocol to improve PA to reduce disability incidence.

## 1. Introduction

In the latest years, general surgery and colorectal surgery are changing their rules and approaches in order to follow the increasing elderly population [1]. A new paradigm of approach may allow to develop new treatment strategies for these patients [2,3]. The burden of comorbidities and frailty expose elderly patients to an inter-individual variability that should be considered and managed [4,5,6]. Moreover, it has been demonstrated how admission to surgical units is responsible for lower physical and mental activity at 30 days after hospitalization [7].

Several studies showed the impact of Physical activity (PA) on brain functions and mental health, resulting in reduced cognitive dysfunctions and mortality after surgery [8,9].

Despite the critical role, the magnitude of PA is currently underestimated in elderly surgical population, in consideration of the relevant physical stress burden [10,11,12].

The “Physical Activity Scale for the Elderly” (PASE) questionnaire is a widely used tool for physical activity assessment in people over 65 years. It was developed on 13 queries, providing questions on spare time, house-work, and work-related jobs [13,14,15,16].

Based on the association between a normal to high exercise ability and clinically relevant benefit in physical activity after hospitalization in surgical units, we decided to use the PASE scale in order to investigate (1) the association of PA with post-hospital stay disability using the activity of daily living (ADL) and the instrumental activity of daily living (IADL) scale, (2) the association of PA with the risk of post-hospital stay immobilization using the Exton-Smith scale [17,18].

Diverticular disease is a very common findings in Western Countries, its presentations arise from bleeding and symptomatic diverticulitis to acute abdomen due to perforation [19]. As depicted in literature, especially for elderly and frail people low and mild acute diverticulitis undergo only conservative treatment to minimize the risk of complications, however hospitalization may aggravate disability [20].

The aim of our study is to demonstrate the impact of PA on the disability risk in elderly population hospitalized for mild acute diverticulitis and diverticular bleeding undergone only conservative management.

## 2. Materials and Methods

### 2.1. Study Population

We retrospectively analyzed data collected as appendix of clinical folders, of all elderly patients admitted to General Surgery Unit of “Antonio Cardarelli” hospital, Campobasso, Italy, undergoing hospitalization for diverticular bleeding and low/mild acute diverticulitis classified as Hinchey I-IIa. Data were collected anonymously. All patients signed a proper informed consent. All patients older than 65 years are screened at the moment of admission and first check up visit using always the same questionnaires (ADL-IADL-Exon-Smith, PASE), which are retrospectively reviewed case by case. Population baseline characteristics are depicted in Table 1.

Interventions are shown in Table 2. Data were prospectively collected from October 2018 to March 2020. Inclusion criteria were: (1) age 65 years and older (2) good daily living and instrumental activities daily living independence, at admission as evaluated by an ADL and IADL score (3) low development risk of pressure injuries evaluated by an Exton-Smith Scale (more than 15 points), (4) patients who underwent to conservative procedures.

We excluded from the study all patients who were addressed to critical care or emergency surgery due to the impossibility to complete the geriatric assessment during pre-screening.

All patients who required surgical, endoscopic or radiological intervention were excluded, because surgical stress should be a potential bias.

We excluded patients affected by baseline delirium, physician-diagnosed dementia, Mini-Mental State Examination (MMSE) score below 24 points [21]. Patients affected by severe auditory or visual deficits were also excluded.

The study was in compliance with the declaration of Helsinki and was approved by our institutional review board (protocol number 06/21, approved date: 10 February 2021). All patients signed a proper informed consent (Appendix A).

### 2.2. Geriatric Evaluation

The same day of hospitalization after admission and clinical stabilization and at the first post-discharge checkup visit (7 to 10 days after discharge), trained physicians completed the ALD, the IADL questionnaire and the Exton-Smith Score to screen patient’s independence status. The first checkup visit were scheduled following patients needs and possibilities.

A short portable mini-mental scale (SPMM) was performed in order to asses cognitive performance [22]. These scales are common during geriatric evaluation and described elsewhere [13]. Moreover, we evaluated physical activity through PASE scale, which is a validated 12 items score designed to measure physical activity of elderly in everyday life. The scale considers walking, exercise, housework, yard work, and caregiving needs (Appendix A) [15].

### 2.3. Statistical Analysis

Continues variables were expressed as mean ± standard deviation (SD), while categorical variables are expressed as number and percentage and compared using the χ2 test. The Bonferroni ANOVA test with post-hoc analysis was used for multiple comparisons. Logistic regression analysis was performed to evaluate the association between PASE and post-hospitalization disability development. Our analysis was corrected for age, gender and BMI. *p* values < 0.05 were considered statistically significant. Analysis was performed using the STATA 11.2 software (Stata Corp. LP, Collage Station, TX, USA).

## 3. Results

We enrolled 56 elderly patients. Mean age was of 75.9 ± 8.9 years old; 57.1% were males. Population baseline characteristic are shown in Table 1. Mean hospital stay was 9.5 days (range 6–13). We found a 30.4% of patients who presented a good physical activity level, 46.4% presented moderate physical activity and 23.2% presented low physical activity evaluated by PASE [23].

Population baseline characteristics were divided in three groups as presented in Table 3. As depicted in the table the oldest patients showed a worse physical activity, but this trend does not reach a statistical significance (*p* = 0.067).

BMI had a good impact on physical activity. Our results showed that a higher BMI is associated with more physical active patients (*p* = 0.044). A better renal function follows the same trend. On the other hand, we observed a loss of activity in patients who had previous underwent oncologic surgery.

Regarding geriatric evaluations we compared pressure injury development and immobilization to ADL, IADL and Exton-Smith Scale. We consider cognitive performance reduction with a cut off of loss of more than 5 points in the Short Port of ADL and IADL and a loss of more than 15 points showed in the Exton-Smith Scale, (*p*-value 0.017 and 0.010, respectively).

We performed a logistic regression analysis to evaluate the potential association of PA with disability after recovery. We also included in the analysis, age, gender, PASE and BMI. PASE did not reach statistical significance, but we found an Odds Ratio of 0.51 95% CI 0.25–1.03 *p* value 0.062 (Table 4), this founding may underline an interesting trend.

## 4. Discussion

Frailty as a result of multi-comorbidities, physical impairment, reduced functional reserve predisposes to increased disability and vulnerability in older adults. This condition could be associated to a variety of post-hospitalization complications, delayed discharge causing poor outcomes and higher costs for the health system [24,25].

It is already well known that physical activity is an important component enhancing the recovery after surgical admission. Kehlet et al. demonstrated that early recovery reduces incidence of the most common complications as thrombo-embolism, delirium, and pneumonia [26].

In this study we analyzed how poor physical activity in elderly population, measured by PASE score, undergoing conservative treatment of mild acute diverticulitis is significantly associated with the development of post-recovery disability. On the other hand, a good physical performance may help elderly patients’ rehabilitation after hospitalization.

We have to underline that a General Surgery Unit in a small region of our country, shall take care of even more elderly patients, who are often alone and poor in literacy, with a severe risk of disability. For that reason, peripheral General Units often are obliged to hospitalize fragile patients to best manage also low and mild acute diverticulitis. At the same time surgeons and territorial medicine should find the best approach to reduce disability incidence before and after hospital stay.

Strong evidences reported how reduced pre-operative physical performance is associated to increased risk of post-recovery complications. Indeed Reilly et al. found that self-reported poor exercise tolerance was associated to an increased risk of post-hospitalization myocardial ischemia, other cardiovascular and neurological events [27]. Subsequently, physical performance might represent a treatable treat in elderly patient undergoing elective surgery. Previous studies documented that preoperative rehabilitation is an effective tool in reducing surgical peri and post-operative risks in both hospitalized or home based patients [28].

In our study we observed a significant association between pre-admission self-reported physical performance and independence status, cognitive performance and mobilization. A recent cohort study revealed that physical activity was associated with reduced risk of post-operative delirium [7] and this association was stronger and more evident among female gender. We observed a significant association between pre-admission physical activity and post-discharge reduction of cognitive performance. In order to avoid confounders, patients with cognitive disorder or a reduced baseline cognitive performance were not considered for the study. Many experimental models reported an increased serum level of brain derivate neurotropic factor (BDNF) after physical exercise and this may suggest a beneficial role of exercise in neurogenesis process [29].

Another important key-point of our findings was the association between low self- reported physical activity linked to lower BMI. Low BMI might be related to reduction of total weight, muscle mass and probably sarcopenia. So, we shall underline the potential bidirectional relationship between sarcopenia and physical exercise, but it would be difficult to explain a cause/effect relationship [30]. Furthermore, enhanced systemic inflammation associated to both sarcopenia and reduced physical exercise may trigger unbalance in adipokines production possibly resulting in worse clinical outcomes [31,32].

On the surgical point of view all patients underwent to different procedures classified only as conservative, because surgical approach should be an important bias related to our aim. Logistic regression analysis performed in our study do not demonstrate that physical activity measured by PASE scale and corrected by gender, age and BMI has independent role on disability after hospitalization, however this result may be underestimated by the limited number of participants in our study.

Based on our data we strongly suggest identifying elderly sedentary patients to better program and asses surgical and non-surgical treatments. In our study, we excluded patients already affected by poor mobility or disabled, because poor physical performance was an important risk factor to develop disability. The goal of our study is to demonstrate how important might be a simple bedside evaluation of patients to identify elderly vulnerable patients. Multidimensional assessment in elderly patients need to balance the intervention-related risks, the functional reserve and post-discharge perspective in order to identify treatable traits and limit the treatment related complications [33,34,35,36].

Finally, the physical dysfunction should be carefully managed among multidisciplinary teams to achieve better outcomes in elderly patient.

### Study Limitations

This is an observational study, single-centered with a limited number of participants and our data should be confirmed in future larger studies.

## 5. Conclusions

Reduced physical activity in everyday life, as indicated by PASE score, in elderly patients is associated with increased post-recovery disability regarding independence, cognitive performance and immobilization. Pre-admission poor physical performance diagnosis may allow to perform a standardized multidimensional protocol to improve PA to reduce disability incidence.

## Figures and Tables

**Table 1 medicina-57-00360-t001:** Overall Population Characteristics.

Characteristics	Overall Population *N* = 56
Age, mean ± SD	75.91 ± 8.91
Gender, males, *n* (%)	32 (57.1)
BMI, mean ± SD	25.39 ± 0.82
Arterial hypertension, *n* (%)	32 (58.2)
Atrial Fibrillation, *n* (%)	7 (12.5)
Diabetes Mellitus, *n* (%)	12 (21.4)
COPD, *n* (%)	8 (14.2)
Drugs Number	3.45 ± 2.67
Previous Neoplastic Surgery	17 (30.3)
Hemoglobin	12.39 ± 2.28 g/dL
Creatinine Hospitalization 9, 5 days (6–13)	0.97 ± 0.49 mg/dL

SD: standard Deviation, BMI: Body Mass Index, COPD: Chronic Obstructive Pulmonary Disease.

**Table 2 medicina-57-00360-t002:** Types of intervention.

Treatment	*N* of Cases	Diagnosis at Admission
DRUG THERAPY	27 patients	Hinchey I-IIa
PERCUTANEOUS DRAINAGE	16 patients	Hinchey IIa-IIb
ENDOSCOPIC TREATMENT FOR BLEEDING	13 patients	Rectal Bleeding

**Table 3 medicina-57-00360-t003:** Population Characteristics Regarding Physical Activity Level.

Characteristics	PASE > 90 *n* = 17	PASE 41–90 *n* = 26	PASE ≤ 40 *n* = 13	*p*-Value
Age, mean ± SD	71.7 ± 6.3	77.9 ± 8.2	77.3 ± 11.5	0.067
Gender, male, n (%)	11 (64.7)	14 (53.8)	7 (53.8)	0.779
BMI	27.05 ± 7.15	25.91 ± 3.63	21.36 ± 7.27	**0.044**
Haemoglobin (g/dL)	12.62 ± 1.74	12.19 ± 2.25	12.49 ± 3.02	0.826
Creatinine (mg/dL)	0.81 ± 1.64	0.91 ± 0.29	1.31 ± 0.84	**0.013**
ADL	5.82 ± 0.39	5.57 ± 1.06	3.91 ± 2.90	**0.038**
Drugs Number	2.52 ± 2.5	3.65 ± 2.48	4.33 ± 3.14	0.178
Previous Neoplastic Surgery	4 (23.5)	5 (19.2)	8 (61.5)	**0.015**
IADL	7.35 ± 1.22	6.0 ± 2.49	3.83 ± 3.48	**0.017**
SPMM < 5 pts, n (%)	1 (6)	5 (19.2)	6 (46.2)	**0.017**
Exton-Smith < 15, n (%)	1 (6)	4 (3.8)	6 (46.2)	**0.010**

BMI: Body Mass Index; PASE: Physical Activity Scale for the Elderly, ADL: Activities of Daily Living; IADL: Instrumental Activities of Daily Living; SPMM: Short Portable Mini Mental Scale. One-way ANOVA with Bonferroni correction was performed.

**Table 4 medicina-57-00360-t004:** Logistic Regression Analysis.

Variable	Odds Ratio	95% CI	*p*-Value
Age	1.05	0.97–1.13	0.177
Gender	1.57	0.45–5.45	0.473
BMI	0.82	0.42–1.61	0.576
PASE	0.51	0.25–1.03	**0.062**

CI: Confidence Interval.

## Data Availability

The datasets used and/or analysed during the current study are available from the corresponding author on reasonable request.

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
