# Peer review of "Impact of Physical Activity on Disability Risk in Elderly Patients Hospitalized for Mild Acute Diverticulitis and Diverticular Bleeding Undergone Conservative Management"

_medicina, 2021, doi:10.3390/medicina57040360_

Round 1

Reviewer 1 Report

All the suggested changes have been included into the last version of the manuscript

I consider that the quality of this paper has a lot improved

Author Response

Dear Reviewer,

Thank you for your support. 

Reviewer 2 Report

The manuscript has been well revised, but I have a few more questions.

  1. It is described in Line 59 that "we decided to use the PASE scale in order to investigate 1) the association of PA with post-hospital stay disability using the We decided to use the PASE scale in order to investigate 1) the association of PA with post-hospital stay disability using the activity of daily living (ADL) and the instrumental activity of daily living (IADL) scale...", thus I can imagine that disability was determined based on the assessment of ADL and IADL at the time of hospitalization and at the first outpatient visit after discharge, but what criteria were used to determine disability is not described. Please describe in the Method which score and at cut-off points (or reduction of the scores?) you defined as disability.

  2.  In some parts of the manuscript, PA and PASE are used synonymously, but when PA, which indicates activity level in a broad sense, is used synonymously with PASE, it is difficult to understand whether it refers to PASE score or to other scores such as ADL and IADL. Please review your word usage. 

  3. In the abstract and text, it says that PASE was scored at admission and at the first outpatient visit, but how each value was used in this study is not described. Of which point PASE score is used in this study? Didn’t this study examine whether PASE at admission was associated with predicting a decline in patient activity over the hospitalization? 

Author Response

  1. It is described in Line 59 that "we decided to use the PASE scale in order to investigate 1) the association of PA with post-hospital stay disability using the We decided to use the PASE scale in order to investigate 1) the association of PA with post-hospital stay disability using the activity of daily living (ADL) and the instrumental activity of daily living (IADL) scale...", thus I can imagine that disability was determined based on the assessment of ADL and IADL at the time of hospitalization and at the first outpatient visit after discharge, but what criteria were used to determine disability is not described. Please describe in the Method which score and at cut-off points (or reduction of the scores?) you defined as disability.

Dear Reviewer thank you for your comments. In our study we perform a logistic regression analysis, because we want to study the impact of Physical Activity at the moment of admission with the reduction of abilities of elderly patients after hospitalization. So it is not important for our aim to define a cut off for disability, but we want to investigate only the role of Physical activity at admission to prevent the cognitive performance reduction after hospitalization. Moreover at line 136 you can find our definition of cognitive performance reduction “with a cut off of loss of more than 5 points in the Short Port of ADL and IADL and a loss of more than 15 points showed in the Exton-Smith Scale “

  1. In some parts of the manuscript, PA and PASE are used synonymously, but when PA, which indicates activity level in a broad sense, is used synonymously with PASE, it is difficult to understand whether it refers to PASE score or to other scores such as ADL and IADL. Please review your word usage. 

In order to solve any misunderstanding we removed the abbreviation PA we write always physical activity.

  1. In the abstract and text, it says that PASE was scored at admission and at the first outpatient visit, but how each value was used in this study is not described. Of which point PASE score is used in this study?

To determine the association between BMI, renal function and other baseline characteristics of patients and PASE we used the admission PASE score. Also admission PASE score was used to try to find a potential association with disability after recovery. We specified it into the text.  Anyway PASE scale and all other  questionnaires are subministrated to patients at admission and at first check up visit  also in order to perform further studies.

  1. Didn’t this study examine whether PASE at admission was associated with predicting a decline in patient activity over the hospitalization? 

We performed, as depicted in methods, a retrospective study, so we can’t perform predictions, but only find relationships.

Moreover, In the results we declare “We performed a logistic regression analysis to evaluate the potential association of Physical Activity with disability after recovery. We also included in the analysis, age, gender, PASE and BMI. PASE did not reach statistical significance, but we found an Odds Ratio of 0.51 95% CI 0.25-1.03 p value 0.062 (Table 4), this founding may underline an interesting trend”

So it is not statistically associated with decline of patient activity over the hospitalization.

Any data were collected during the hospitalization, but only at admission and at first check up visit.

This manuscript is a resubmission of an earlier submission. The following is a list of the peer review reports and author responses from that submission.

Round 1

Reviewer 1 Report

The number of authors seems a little excessive (14). Please can you eliminate some of them and reduce to (10), or even less.

You must define the criteria choosen for defining the term of “elderly population” in the abstract and also the number and gender of the patients included into this study and the period of time and the year in whom was realized and to describe if was prospective in its design.

You must explain the characteristics of the classification of Hinchey grades I, II-a and II-b, and include one reference for describing their characteristics

How may days elapsed from the admission and clinical stabilization until the performance of different questionnaires of ALD, IADL,ESS and PASE?

In the Discussion the authors comment : In our study we observed a significant association between pre-admission self-reported physical   performance and independence status, cognitive performance and mobilization. Although this may be an intuitive association, the impact of physical activity on disability rate after surgical discharge has not been completely clarified.The authors have some doubts about the interpretation of their results.

The authors comment about the Study Limitations : This is an observational study, single center, with a limited number of participants and our data should be confirmed in future larger studies.

There are 5 references not numerated and of course not included into the text

Q1 : Title and Number of Authors

Is complete and very informative. You can suppress “TO” between underwent and conservative

The number of authors seems a little excessive (14). Please can you eliminate some of them and reduce to (10), or even less.

Q2: Abstract and keywords

You must define the criteria choosen for defining the term of “elderly population” in the abstract and also the number and gender of the patients included into this study and the period of time and the year in whom was realized and to describe if was prospective in its design.

In the keywords you can include the terms, Diverticulitis, Mental activity, Physical activity scale of daily living (PASE), Activity of daily living (ADL), the Instrumental activity of daily living (IADL) and the Exton-Smith Scale

Q3. 1. Introduction

Is clearly described and informative about the main aims of the study

Q4. 2. Materials and Methods

2.1. Study Population

Line 63 : It is not clear if this study was prospective or retrospective. Please clarify this important point

Line 69 : In the first pass, you must say “Age 65 years and older ” instead of “age at least 65 years”

You must describe the total number and gender of the patients included and how many were hospitalized by diverticulitis and the other just for bleeding

You must explain the characteristics of the classification of Hinchey grades I, II-a and II-b, and include one reference for describing their characteristics

And also will be convenient to explain how do you obtain the ADL and IADL scores with their references and the same with the Exton-Smith Scale and their references

2.2. Geriatric evaluation

How may days elapsed from the admission and clinical stabilization until the performance of different questionnaires of ALD, IADL,ESS and PASE?.

Please specify the medium time and the minimum and maximum intervals, according to the ages of the participants

2.3. Statistical Analysis

Is well described and OK

Q5.3. Results

Line 101 : You must include only one decimal ,describing the mean age including the standard deviation

Line 102 : You must write “shown” , instead of “showed”

You must describe first the characteristics of the population changing the order of the tables putting the table 2 before the table 1

Instead of writing “Hypertension” you must change for “Arterial hypertension”

In Hemoglobin levels, you must add (g/dl)

In Creatinine levels, you must add (mg/dl)

In Hospitalization you must include days in the left column and eliminate this word from the right one

In Table 2 please write Drug therapy in singular, not plural

“Percutaneous drainage” not “dreinage”

In Table 3, please explain the meaning of PASE in the title

Eliminate the second decimal from the all ages included and

Please, include the units of measurement of Hemoglobin and Creatinine mean values, the ADL and IADL

Line 127 : Please don´t repeat PASE twice. You can change by “The latter or this one…”

Q6. 4. Discussion

The authors comment about their findings is at least, doubtful,l such as they express literally :

“In our study we observed a significant association between pre-admission self-reported physical   performance and independence status, cognitive performance and mobilization. Although this may be an intuitive association, the impact of physical activity on disability rate after surgical discharge has not been completely clarified” ……

Based on our data we strongly suggest to identify elderly sedentary patients to better programme and asses surgical and non-surgical treatments. In our study, we excluded patients already affected by poor mobility or disabled, because poor physical performance was an important risk factor to develop disability …….

I fully agree with this mention

Study Limitations : This is an observational study, single-centred (center) with a limited number of participants and our data should be confirmed in future larger studies.

Q7. Conclusions

This is the most important final message indeed:

Pre-admission poor physical performance diagnosis may allow to perform a standardized multidimensional protocol to improve PA to reduce disability incidence

Q8. References

1.- There is one reference not numbered before the first one. If you want to include it all the references must be  re-numbered again

2.- After the Ref. 2, there is another Ref (Marte G et al) not numbered. If you want to include it all the references must be  re-numbered again

3.- After the Ref. 3, there is another Ref (Lee,ss et al) not numbered.  If you want to include it all the references must be  re-numbered again

4.- After the Ref. 23, there is another Ref (Krause et al) not numbered. If you want to include it all the references must be  re-numbered again

5.- After the Ref. 26, there is another Ref. (Montroni I et al), not numbered. If you want to include it all the references must be  re-numbered again

6.- In total there are 5 references not numbered that you must to include into the text in the correct place and also in the final list of total references

Reviewer 2 Report

In this manuscript, the authors aimed to investigate the predicting risk factor of disability after recovery in elderly patients hospitalized for diverticulitis and diverticular bleeding. However, there are several points to be revised.

  1. Why were patients with diverticulitis and diverticular hemorrhage included in the study? It is decided abruptly in the Method section, but the background of the decision to conduct the study on this subject is not stated.

  1. The title says that the object of study is patients with diverticulitis, but this study also covers diverticular bleeding cases.

  1. According to the Methods, patients with diverticulitis and diverticular bleeding were included, and some of them were excluded according to the exclusion criteria. However, the manuscript only shows the number of cases that were involved in the final analysis. In order to ensure the transparency of the study, it is necessary to present a flow diagram in accordance with STROBE, showing how many patients met the inclusion criteria and how many were excluded and for what reasons.

  1. Were there any cases of diverticular bleeding that were treated conservatively without endoscopic treatment?

  1. Line 58, please provide a reference (Katz, Lawton) for the calculation of the ADL and IADL scores.
    Additionally, please show the reference of SPMM scale in line 87.

  1. Were there any interventions to maintain ADLs such as rehabilitation during the hospitalization? If so, how were they treated in this study? Please describe it in the Methods.

  1. In the Methods, there is no description about the outcome of this study. Please describe clearly what is the outcomes and predictor, and how the values were defined (timing of record, judgement of disability, etc.).

  1. In line 96, it is described that association between PASE and post-surgery disability development was evaluated, but current study did not include surgery cases.

  1. In line 116, the authors described that the relationship between pressure injury development and immobilization and ADL, IADL, and Exton-Smith Scale was compared. However, in Table 3 only shows the relationship between PASE score at admission and changes in ADL, IADL, and Exton-Smith Scale is described, but the relationships between pressure injury development or immobilization and those scales are not shown. It is necessary to explain in the Methods how “pressure injury development” was defined (timing of judgement etc.) and what was analyzed. As a same, what is the definition of “cognitive performance reduction” described in Line 118?

  1. The Method states that each score of ADL, IADL, and Exton-Smith Scale was assessed on admission. However, the Method does not state the definition of the reduction of the scores, which is estimated in line 117 and Table 3. Which timing of the scores did the authors compare? Please describe the definition in the Methods.

  1. Line 125, what is the definition of “PA with disability after recovery”? Please provide criteria for the definition in the Methods.

  1. Why did the authors choose the four covariates of age, gender, BMI, and PASE in the multivariate analysis in Table 4?

  1. In the manuscript and abstract, it should be more clearly stated that the assessment of physical activity at the time of admission is useful in predicting the outcomes at the time of discharge.

  1. In the conclusion of the manuscript, the authors says that PASE is associated with post-recovery independence, cognitive performance, and immobilization. However, the associations between PASE and SPMM or other scores at the time of admission were evaluated (Table 3), but those at the timing of post-recovery were not estimated. Table 4 shows the association between PASE and PA with disability after recovery, but there is no definition of disability in this manuscript, and the evidence of the association between PASE and independence, cognitive performance, and immobilization is not provided.

  1. In the conclusion of the manuscript and abstract, the authors described that “pre-admission poor physical performance diagnosis may allow to perform a standardized multidimensional protocol to improve PA to reduce disability incidence”, but the effect of a standardized multidimensional protocol is not estimated in this study. This is a consideration of the authors, not a conclusion of this study.
